# Structural Thermokinetic Modelling

**DOI:** 10.3390/metabo12050434

**Published:** 2022-05-11

**Authors:** Wolfram Liebermeister

**Affiliations:** Université Paris-Saclay, INRAE, MaIAGE, 78350 Jouy-en-Josas, France; wolfram.liebermeister@inrae.fr

**Keywords:** metabolic model, structural kinetic modelling, dependence schema, reaction elasticity, model ensemble

## Abstract

To translate metabolic networks into dynamic models, the Structural Kinetic Modelling framework (SKM) assumes a given reference state and replaces the reaction elasticities in this state by random numbers. A new variant, called Structural Thermokinetic Modelling (STM), accounts for reversible reactions and thermodynamics. STM relies on a dependence schema in which some basic variables are sampled, fitted to data, or optimised, while all other variables can be easily computed. Correlated elasticities follow from enzyme saturation values and thermodynamic forces, which are physically independent. Probability distributions in the dependence schema define a model ensemble, which allows for probabilistic predictions even if data are scarce. STM highlights the importance of variabilities, dependencies, and covariances of biological variables. By varying network structure, fluxes, thermodynamic forces, regulation, or types of rate laws, the effects of these model features can be assessed. By choosing the basic variables, metabolic networks can be converted into kinetic models with consistent reversible rate laws. Metabolic control coefficients obtained from these models can tell us about metabolic dynamics, including responses and optimal adaptations to perturbations, enzyme synergies and metabolite correlations, as well as metabolic fluctuations arising from chemical noise. To showcase STM, I study metabolic control, metabolic fluctuations, and enzyme synergies, and how they are shaped by thermodynamic forces. Considering thermodynamics can improve predictions of flux control, enzyme synergies, correlated flux and metabolite variations, and the emergence and propagation of metabolic noise.

## 1. Introduction

In this article, I describe a modelling framework called Structural Thermokinetic Modelling (SKM, [1,2]) in which metabolic models are constructed around given metabolic states. STM extends the existing Structural Kinetic Modelling paradigm (SKM) to account for reversible reactions and thermodynamic forces. This makes it a good tool for assessing the effects of these forces on model dynamics. Furthermore, for metabolic systems with unknown parameters, STM provides a convenient and flexible framework for model fitting, ensemble modelling, and optimisation that keeps track of uncertainties while imposing constraints set be the modeller.

Metabolic fluxes in cells are shaped by network structure, reaction thermodynamics, enzymatic rate laws, and enzyme regulation. Flux, metabolite, and protein data provide a detailed picture of metabolism, and computational models can help us answer important biological questions. For example, how do enzyme-inhibiting drugs, enzyme overexpression, or changes in nutrient levels affect a cell’s metabolic state? Will local enzyme perturbations have long-range effects on the fluxes or are they compensated by changes of nearby metabolite concentrations [3]? To address such questions, metabolic networks, comprising thousands of reactions, have been built [4,5], and methods such as Flux Balance Analysis (FBA) [6,7], MoMA [8], the principle of minimal fluxes [9], or ROOM [10] predict plausible flux distributions from heuristic assumptions [11] or sampling [12]. Thermodynamic flux analysis [13,14,15,16,17] relates fluxes to metabolite concentrations via equilibrium constants and thermodynamic forces. However, it does not describe how metabolic fluxes arise mechanistically. How can we predict the effects of enzyme concentrations, external metabolite concentrations, or parameters like temperature or the dilution rate on fluxes? If metabolite concentrations were constant, reaction fluxes would be directly proportional to enzyme concentrations. However, this is not the case in reality: the interplay between fluxes, concentrations, and enzyme activities leads to stationary fluxes and metabolite concentrations which depend on enzyme levels in complicated ways. To understand how enzyme inhibition changes metabolic fluxes, we need to know its effects on metabolite concentrations. Flux analysis cannot describe this because it does not consider rate laws, enzyme saturation, and regulation by effector molecules. Kinetic models would tell us how enzyme perturbations change the fluxes and metabolite concentrations in steady state, but kinetic rate laws and rate constants are largely unknown, especially in less well-studied organisms.

Close to a steady state, metabolic dynamics can be approximated by linear models. Metabolic Control Theory [18,19,20] (MCT, Figure 1) considers perturbations such as enzyme inhibitions, quantifies their effect on reaction rates, and infers network-wide steady state changes. MCT uses two types of sensitivity coefficients: reaction elasticities describe how reaction rates change with changing metabolite concentrations: they concern immediate effects on a fast time scale and depend only on the rate law of the perturbed reaction. Response and control coefficients, in contrast, describe long-term, wide-range effects of perturbations on steady-state fluxes and concentrations (see Appendix A). MCT has various applications. Using response coefficients we can assess the effects of enzyme inhibition, differential enzyme expression, varying external metabolite concentrations, enzyme inhibition by drugs, or genetic modifications. A Taylor expansion based on response coefficients can also describe complicated perturbations (e.g., activity changes of all enzymes), optimal enzyme allocation in pathways [21], and parameter uncertainty or variability [22]. Second-order response coefficients [23], describing synergy effects, play a role in predicting optimal differential expression [24]. In systems with periodic [25,26] or random [22] parameter fluctuations, the variations of fluxes and metabolite concentrations depend on spectral response coefficients. Importantly, all these phenomena can be described without a full kinetic model: linearised models with the same network structure and reaction elasticities as in the original model suffice to describe the dynamics of small fluctuations caused by static or periodic parameter perturbations or chemical noise, and powerful theory exists for optimal control and model reduction in such linear models [27]. Thus, all we need to know are network structure and elasticities. However, if only the network structure is known, how can we obtain elasticities from few or no other data?

Structural Kinetic Modelling, the ORACLE framework [28,29,30,31,32], and similar methods [33,34,35,36] replace reaction elasticities with random numbers. Reaction rates increase with the substrate concentrations and decrease with the product concentrations. This is encoded in the signs of reaction elasticities: they are positive for substrates and activators, and negative for products and inhibitors (where “substrate” and “product” refer to the flux directions). Assuming irreversible rate laws, SKM equates the scaled elasticities to saturation values, numbers between 0 and 1 that describe the fraction of catalysing enzyme molecules that are bound, on average, to metabolites. In SKM, saturation values are treated as free variables and sampled from random distributions. Possible ranges can be derived from rate laws, e.g., a range ]0,1[ for substrate elasticities in mass-action kinetics. The resulting elasticity matrix is sparse, reflecting network structure and regulation arrows. Each row describes one reaction: substrates and activators lead to positive matrix elements, while products and inhibitors lead to negative elements. To sample the elasticities, each non-zero matrix element is replaced by a random number with the required sign. A sampled elasticity matrix defines a linearised kinetic model, whose Jacobian matrix determines the stability of the reference state as well as the dynamic behaviour around it. Given metabolite concentrations and fluxes, reaction elasticities can be converted into kinetic constants: hence, instead of sampling elasticities, one may also sample some kinetic parameters directly and compute the others [37,38]. Structural kinetic modelling has been applied to various cell biological questions [39,40,41,42].

If elasticities are variable, uncertain, or unknown, what variability or subjective uncertainty can we expect in a model? To explore this, we can study model ensembles, models in which some model features are given (e.g., network structure and flux distribution), while others are varied (e.g., metabolite concentrations and kinetic constants). By sampling variable features from random distributions [22,43] and translating the linearised model back into a full kinetic model, a model ensemble can be created. Given a set of sampled model instances, we can estimate the probability distributions for an infinitely large ensemble. In practice, by generating many model instances (with the same network structure, but different elasticities), one can explore the dynamics allowed by our network and assess their probabilities. By computing the probabilities for different model outputs or types of behaviour, we can see how they depend on network structure, fluxes, or rate laws. If most of the model instances show a certain behaviour, this behaviour can be attributed to the model features kept fixed, while varying properties may be attributed to the features sampled. In a model ensemble, we can also screen for models with a given property—e.g., stable oscillations—and check which model details—such as inhibition arrows or specific elasticity values—are overabundant in this subensemble, and thus potentially responsible for these features [1]. Finally, different model variants can be compared: each variant is translated into a model ensemble and significant differences between the ensembles can be attributed to differences in model structure.

Despite all its merits, ensemble modelling by SKM has one major drawback: it ignores the fact that elasticities are interdependent due to basic physical laws. The net flux in chemical reactions results from a difference of one-way rates $v_{+}$ and $v_{-}$, the rates of microscopic reaction events in forward and backward direction. In chemical equilibrium, the ratio of product and substrate concentrations is given by the equilibrium constant, the net rate *v* vanishes, and the two one-way rates must be equal. More generally, their ratio is given by $\frac{v_{+}}{v_{-}}=e^{\theta }$, where the thermodynamic force $\theta =-\Delta_{r}G/RT$ represents the reaction Gibbs free energy Δ_r_*G*, but dimensionless and with a flipped sign [44]. The quantity $A=-\Delta_{r}G=RT\theta$, still dependent on Boltzmann’s gas constant *R* and absolute temperature *T*, is called reaction affinity. The thermodynamic force θ depends on the chemical potentials and therefore on substrate and product concentrations, and can range between two extremes. In equilibrium reactions, the force vanishes, the one-way rates $v_{+}$ and $v_{-}$ are equal, and the net flux *v* vanishes too. Strongly driven reactions, in contrast, have large thermodynamic forces, a negligible backward flux $v_{-}$, and a net flux close to the forward flux $v_{+}$. By tuning the one-way fluxes ($v_{+}=\frac{e^{\theta }}{e^{\theta }-1}v,\frac{1}{e^{\theta }-1}v_{-}=v$), the thermodynamic force shapes elasticities: if a force is large, the backward rate will be negligible, the net rate does not depend on the product concentrations, and the product elasticities vanish. Since the forces themselves are subject to Wegscheider conditions [45] (they must sum to zero over loops in the network) and elasticities depend on them, elasticities may be interdependent across the entire network [46]. Ignoring this fact, SKM leads to thermodynamically inconsistent models [46] (for an example, see Appendix A).

How can we solve this problem? Sampling correlated elasticities—satisfying network-wide thermodynamic constraints—seems difficult, but it is actually easy for certain rate laws: using formulae from [46], elasticities can be computed from thermodynamic forces and saturation values, both of which can be independently varied. Below, I use this to construct model ensembles: I describe an algorithm for sampling reaction elasticities, while fully accounting for dependencies due to reversible rate laws. I then show how thermodynamics and enzyme saturation shape control properties, dynamic time courses, enzyme synergies, and fluctuations in metabolic systems. Importantly, also enzyme synergies and other second-order effects are described by closed formulae.

Since it adds thermodynamics to SKM, the framework is called Structural Thermokinetic Modelling (STM). A thermodynamically feasible metabolic state serves as a basis for constructing thermodynamically feasible models. Compared to the original SKM, the modelling procedure is more formalised: all model variables (including metabolite concentrations and fluxes) are determined step by step by inserting known values, sampling, or optimisation. The workflow follows a schema that describes the model variables, guarantees physical correctness, and can be used to define probability distributions. By separating network (defining the biological system) and schema (defining types of variables and their physical dependencies), models can be built flexibly, and their mathematical relationships are easy to see. This makes STM well-suited for automatic model construction.

Studying thermodynamic forces does not only lead to more realistic models, but can also reveal important aspects of cell physiology. The forces do not only determine possible flux directions, but have also effects on reaction rates at given enzyme levels, enzyme demands at given desired fluxes, the reversibility of reactions, the controllability of fluxes, the generation of chemical noise, and its propagation across the network. Since all this affects the proper functioning of cells, the thermodynamic forces in a metabolic network will not be distributed randomly, but are well adjusted to the flux profile. Thermodynamic forces should not be too small because this would reduce enzyme efficiency and create thermodynamic bottlenecks. Force patterns avoiding such bottlenecks can be computed [47]. In contrast, large driving forces entail a energy high dissipation which reduces the yield on substrate. However, aside from this trade-off, placing high or low driving forces in specific reactions in the network can have favourable effects. The first reactions in unbranched biosynthesis pathways are often strongly forward-driven, which is expected to improve a pathway’s dynamic control properties [48]. Theoretically, strongly forward driven reactions can serve as “diodes” for fluctuations: metabolic noise and, possibly, information-carrying signals pass only in one direction. Moreover, in substrates of thermodynamically unfavourable reactions, the concentration tends to increase with the flux, which makes these metabolites suitable flux sensors [49].

While hypotheses about force patterns and their functions are easily made, testing such claims is difficult because the forces cannot be tuned independently: emerging from metabolite concentrations, the forces in a network are under many constraints. Moreover, direct effects of forces may be hard to distinguish from effects of the underlying metabolite concentrations: for example, a high product level will decrease a reaction rate in two ways: by decreasing the driving force and by saturating the enzyme. In the rate law, these effects appear entangled and disentangling them is not easy: if we vary an enzyme concentration experimentally, this changes metabolite levels, fluxes, thermodynamic forces, and enzyme saturation levels at once. STM enables targeted changes in models, for example, varying the thermodynamic forces while keeping all fluxes unchanged and averaging over many possible choices of saturation values. By assessing the results, we can study the effects of thermodynamic forces alone, which casts light on their possible functional roles.

## 2. Results

### 2.1. Structural Thermokinetic Modelling

Structural Thermokinetic Modelling (STM) is a framework for building kinetic metabolic models with reversible rate laws. It is simple, can flexibly integrate available data, yields consistent models, and can be used for semi-automatic model construction. STM is based on a dependence schema, describing the dependencies between variables by linear or nonlinear functions. In contrast to SKM, elasticities are not directly given by saturation values but depend additionally on thermodynamic forces. Elasticities can be computed from thermodynamic forces and saturation values for a number of rate laws [46]. The resulting elasticity matrices reflect network structure, fluxes, thermodynamics, and enzyme saturation with reactants and regulators. Any choice of basic variables leads to consistent models and any consistent model can be obtained by a choice of the basic variables. The dependence schema can also be used for error propagation or for tracing small perturbations: in this case, arrows in the schema correspond to connection matrices, containing derivatives between model variables. STM can also be used to define probability distributions: a probability distribution of the independent basic variables determines the distributions and correlations of all variables. In turn, any feasible distributions of model variables can be defined by distributions of the basic variables.

To construct metabolic states and kinetic models, STM follows the dependence schema (Figure 2): basic variables are freely chosen, while derived variables are computed from them. In practice, “choosing” can mean that variables are sampled, chosen manually, fitted to data, or optimised. During the stepwise model construction, various pieces of data can be included. In the “metabolic state” phase, we choose thermodynamically feasible fluxes, metabolite concentrations, and equilibrium constants (which in turn may depend on Gibbs free energies of formation or independent $k^{\mathrm{eq}}$ values as basic variables). The flux distribution must be thermodynamically feasible: flux directions must follow the thermodynamic forces, which depend on metabolite concentrations and equilibrium constants. All these variables can be obtained by existing methods for thermodynamic flux modelling: for example, equilibrium constants can be obtained by eQuilibrator [50] and feasible metabolite concentrations can be obtained by MDF [47]. In the “kinetics” phase, saturation values are chosen in the range ]0,1[ (or possibly in a smaller range or using a beta distribution) and the elasticity matrix is computed using Equation (Equation 1). Each elasticity matrix corresponds to a consistent kinetic model with reversible rate laws which can be easily reconstructed. From the elasticity matrix, we further obtain the unscaled elasticities $E_{c_{i}}^{v_{l}}$, the Jacobian matrix and response or control matrices used in Metabolic Control Theory. Using these matrices, we can study model properties such as dynamic stability, oscillations, linearised temporal dynamics, and the propagation of noise. Finally, without any further sampling, the second-order elasticities can be computed. From the first and second order elasticities, we obtain second-order response and control coefficients, which describe synergy effects of double parameter perturbations.

Figure 3 shows how a model of central metabolism in *Escherichia coli* can be constructed step by step. The model is a modified version of the model from [51], with additional exchange reactions for some of the biomass precursors. The stationary flux distribution was obtained from measured metabolic fluxes by projecting flux data onto the space of stationary fluxes while constraining the flux directions. Metabolite concentrations and thermodynamic forces for the reference state were determined by balancing [52] metabolite concentration and reaction Gibbs free energy data, in agreement with flux directions. Model structure and data were taken from [51] (for details, see Data Availability). The reference state satisfies all relevant constraints: stationary fluxes, Wegscheider conditions for equilibrium constants and thermodynamic forces, Haldane relationships for kinetic constants, and consistent directions of fluxes and driving forces. For the saturation values, all enzymes were assumed to be half-saturated, setting all saturation values to standard values of $\frac{1}{2}$ (for the choice of this value, see Appendix A). Instead of assuming the same saturation value for all enzymes, we may also insert known, specific saturation values (obtained from known *k*^M^ values and metabolite concentrations); for ensemble modelling, we may sample them in the range ]0,1[ or around plausible values. In the model, small-molecule regulation of enzymes (such as allosteric inhibition) was ignored, but it could easily be included.

Will models constructed by STM be biologically plausible? To test this, I considered a model of the threonine pathway in *E. coli* [55] (from BioModels Database [56]) and constructed a twin version with the same reference state and elasticities set to $\frac{1}{2}$. To compare the two models, I simulated a sudden concentration increase of the external substrate aspartate (see Appendix A). Despite the different kinetic constants, the simulations show similar qualitative dynamics and time scales. Hence, realistic fluxes and metabolite concentrations, together with plausible assumptions about enzyme saturation, can suffice to obtain realistic unscaled elasticities and a plausible dynamic behaviour.

Aside from constructing a single model, we can also build model ensembles with broad parameter distributions. Model ensembles can be used to study the effects of model structure, flux distribution, thermodynamics, or enzyme saturation on metabolic behaviour, and to test which details have significant effects on model outputs. For example, to study the role of thermodynamic forces in flux control, we may generate two model ensembles with the same fluxes, but different forces. Within each ensemble, saturation values are varied and flux control coefficients are computed. By comparing the two ensembles, we can check which control coefficients differ significantly due to the different forces. To assess significance for a single control coefficient, its distributions (arising from sampled saturation values) are compared between the ensembles (representing different choices of forces). Since a model contains many different control coefficients, the problem of multiple testing must be addressed [57] (see Methods and Appendix A).

Ensemble models can answer a wide range of questions, e.g., how metabolic dynamics, homeostasis, and control depend on network structure, thermodynamics, enzyme saturation, or regulation. Even with little data, we can study control or synergy coefficients, compute their distributions and correlations, and see which control coefficients differ significantly from zero. All these predictions are probabilistic, reflecting the uncertainties arising due to missing or imprecise data. While building a model ensemble does not require any kinetic data at all, inserting some data (e.g., kinetic constants) will decrease variability and narrow down the model results. Hence, data that would usually not suffice for model fitting can still be used for model predictions by STM, where their uncertainty ranges can be assessed. The basic STM approach can be extended in various ways. Appendix A describes “biological” extensions considering cell compartments, metabolite dilution by cell growth, models with thermodynamically infeasible fluxes, divergencies close to chemical equilibrium, enzymatic reactions composed of elementary steps, multiple steady states, and adaptations of enzyme levels. It also describes some statistical extensions regarding prior distributions for saturation values, the analysis of sampled target variables, significant differences between model variants, and ways to impose distributions of target variables.

In summary, STM helps us build realistic metabolic models, study their control properties, and assess how these properties vary. The underlying dependence schema shows how model variables depend on each other. Equipped with these methods, we now study the role of thermodynamic forces in metabolic dynamics and control. In the following sections, I show how different aspects of metabolic dynamics—including flux control, linearised metabolic dynamics, enzyme synergisms, and metabolic fluctuations—are shaped by the pattern of thermodynamic forces in the network.

### 2.2. Metabolic Effects of Gene Expression Changes

When enzyme levels are changing, how will this change the network-wide fluxes? In Flux Balance Analysis (FBA), these effects can be modelled by changing the flux bounds (as a proxy for changing enzyme activities) and re-optimising the fluxes. In MCT, the effects of enzyme changes are directly described by control and response coefficients, and synergy coefficients can be additionally used for more precise approximations [22] or to account for the adaptation of other enzymes [24].

If a target variable such as ATP or biomass production contributes to cell fitness, enzymes should have a positive effect on this variable: if an enzyme had a negative effect, the cell would benefit from downregulating this enzyme [58] and would probably have done so already! In enzyme-optimal states [21,59], the marginal cost and benefit of each enzyme must be balanced, so each enzyme must have a positive control over the metabolic objective, i.e., a positive marginal benefit [59]. Likewise, in FBA models, an enzyme knock-down can decrease the metabolic benefit, but can never increase it—otherwise, FBA would have chosen a smaller flux from the start. However, this holds only if we assume optimal states. Generally, without optimality assumption, enzymes can have positive or negative control over different fluxes: for example, inducing some enzymes may reduce biomass production.

To see which enzymes are likely to have a negative flux control over a flux objective, I built a series of models describing the flux distributions in central metabolism of human hepatocytes. Starting from the large Hepatonet1 network, sparse flux distributions for specific objectives (for example, ATP regeneration during aerobic growth on glucose) were determined by FBA with flux minimisation. In the original paper [9], for each of these flux distributions, a network model was built by omitting reactions with inactive fluxes. I adopted these models and applied STM as described for the *E. coli* model. Metabolite concentrations were determined by thermodynamic balancing of measured concentrations in *E. coli* as a substitute for human hepatocyte data. Using these flux distributions, I obtained models representing a large number of different flux objectives and checked their control coefficients. For example, with ATP production as the flux objective (details in Appendix A), all active enzymes have a positive control over ATP production. This was a typical case: usually, most of the active enzymes had a positive control over the flux objective—even if the models were not constructed to be in enzyme-optimal states! Apparently, suppressing unnecessary fluxes in the FBA step, based on a flux objective, already led to a state in which most enzymes have a positive control over the flux target once kinetics are considered. While this makes intuitively sense, it provides strong support for FBA: even if FBA ignores kinetics, it seems to provide a good starting point for kinetic models, providing fluxes that are likely to support enzyme-optimal states.

While enzymes can have a negative flux control (e.g., a control over reactions that compete with the enzyme for substrate), it is hard to imagine that they negatively control their own flux. However, there are examples of this. Figure 4a shows results from a variant of our hepatocyte model, but with UTP production during anaerobic growth on glucose. UTP is regenerated from UDP via the phosphotransferase reaction UDP + ATP ↔ UTP + ADP, catalysed by nucleoside diphosphate kinase (NDK). The influence of NDK on its own steady-state flux is described by the flux control coefficient $C_{e_{\mathrm{UTP}}}^{v_{\mathrm{UTP}}}$. Surprisingly, in a kinetic model constructed by STM with half-saturated enzymes, this control is negative: a higher enzyme level decreases the flux! This paradoxical effect also shows up in dynamic simulations: when the enzyme level increases, the flux first increases as well, but then drops below the original flux. If we plot the steady-state flux against the enzyme concentration, the slope of the curve—i.e., the response coefficient $R_{e_{\mathrm{UTP}}}^{v_{\mathrm{UTP}}}$—is negative in the reference state. Probably, the reason is that UTP production consumes ATP; due to the turbo design of glycolysis [60], a high ATP consumption reduces the ATP level drastically, entailing a decrease in UTP rephosphorylation. In contrast, a lower NDK level allows ATP to recover, and UTP rephosphorylation increases. Is this paradoxical self-inhibition behaviour typical or is it a rare exception, maybe caused by our choice of identical saturation values of $\frac{1}{2}$?

To see this, we can use STM: with saturation values randomly drawn between 0 and 1, about 90 percent of the models show the paradoxical self-repression effect. Thus, this effect does not depend on fine-tuned parameters, but is likely given our network structure, flux distribution, and metabolite concentrations. To further assess the role of the thermodynamic forces, I varied them proportionally (see Appendix A): when bringing all reactions closer to equilibrium, the effective self-inhibition stopped, while increasing the driving force increased self-inhibition. The example shows how STM can help us understand how network structure, reference state, thermodynamic forces, or details of enzyme kinetics contribute to metabolic dynamics. How they affect emergent system properties and to what extent; and whether an observed behaviour is surprising or likely, given all our physical and physiological knowledge. I end this section with a question for the interested reader: if enzymes such as NDK can “paradoxically” decrease their own flux, how should such enzymes be regulated?

### 2.3. Enzyme Synergisms and Epistasis

Interactive effects of several enzyme levels (or other parameters) on target variables are called synergisms. In STM, synergy effects are obtained from the second-order response coefficients, also called synergy coefficients (see Figure 3h for the *E. coli* model). While synergisms can be computed for any perturbation parameters, including concentration variations in the growth medium, here we focus on enzyme pairs. How do STM predictions compare to predictions by FBA? Figure 5 shows a comparison between STM and two constraint-based modelling frameworks: classical FBA and MoMA (see Methods). The synergisms predicted by MCT confirm our expectations (see Appendix A): cooperating enzymes (e.g., in the same metabolic pathway) show buffering synergisms (inhibiting one of them impairs the pathway function, and inhibiting the second enzyme has less extra effect), while enzymes in alternative pathways show aggravating synergisms: an inhibited pathway can still be bypassed, and only simultaneous inhibitions take effect.

In constraint-based models, there is no direct formula for synergisms: they need to be computed numerically one by one. The formulae of MCT, in contrast, show how synergisms reflect network structure and flux directions (which also define “upstream” and “downstream” enzymes). The synergy coefficients depend on three factors (formulae in Appendix A): on flux control coefficients $C_{v}^{j}$ between the two catalysed reactions; on control coefficients $C_{u}^{y}$ between these reactions and the target variables; and on second-order elasticities in the entire network. The formula suggests that a large first-order control between perturbed reactions and target variable, as well as a large control between the reactions, increase the positive or negative synergisms. Enzyme synergisms computed by MCT depend on the enzyme perturbed, but not on the perturbation parameter: there is no difference between enzyme knock-outs, enzyme-inhibiting drugs, or other perturbations that decrease enzyme activities. In the predictions, the synergy effects for a double knock-down, a double enzyme inhibition, and a combination of knock-down and enzyme inhibition will be the same. This is in line with experiments: genes with many epistasis partners are also more likely to show gene-drug interactions [61]. In reality, cells may compensate enzyme perturbations by changes of gene expression, which changes the overall synergisms. This effect was not considered here, but our synergy coefficients could be corrected for optimal enzyme adjustments [24]. Importantly, enzyme synergisms are just a specific example: MCT defines synergisms not only between enzyme levels and metabolic fluxes, but between any pairs of model parameters (including external metabolite concentrations) and with respect to any target variables (including metabolic fluxes and concentrations).

### 2.4. Uncertain States and Metabolic Fluctuations

Model ensembles can help us model variability between cells, fluctuations in time (e.g., of metabolite concentrations inside a single cell), and uncertainty in models due to missing or uncertain data. Notably, the variability and correlations of cell variables are shaped by network structure and enzyme kinetics. Metabolite concentrations and fluxes that vary in time can be described by random fluctuations that emerge, for example, from variable enzyme concentrations, external perturbations, or chemical noise and propagate in the network. All sources of noise have their own dynamics. Fluctuating enzyme concentrations, for example, can be caused intrinsically (by noise in transcription and translation) or extrinsically (by variability in other cell variables) [62,63]. In their frequency spectrum, typical frequencies are on the time scale of protein dynamics (protein degradation or dilution in growing cells) or slower. Chemical noise arises from the stochastic rates of single reactions because single reaction events occur randomly. The fluctuations percolate through the network, add up, and cause variability in all metabolite concentrations and fluxes. For an example, see Figure 6 (calculations are explained in Methods). If variations are slow, we can describe the cells by a random ensemble of steady states.

For an example of temporal fluctuations, we consider chemical noise, one possible cause of metabolic fluctuations. Individual reaction events happen randomly, which can be described by random fluctuations in the reaction rates. A typical enzyme (with 1000 copies in a bacterial cell and *k*^cat^ = 10 s^−1^) catalyses around $10^{4}$ net reaction events per second. If we assume strongly driven reactions (i.e., neglect backward rates), count the reaction events within one-second time intervals, and describe the count numbers by a Poisson distribution, we expect $10^{4}$ events per second on average with a standard deviation of $\sqrt{10^{4}}=10^{2}$, i.e., a relative standard deviation of one percent. Over larger time intervals, the relative standard deviation is smaller because fluctuations average out. How does this chemical noise translate into metabolite and flux fluctuations? In metabolism, noise from different reactions propagates through the network, adds up, becomes damped or sometimes amplified, and leads to correlated fluctuations of metabolite concentrations and fluxes. To model this, we can add a white noise term to each rate law, which leads to a chemical Langevin equation [64]. The white noise spectrum contains all frequencies with uniform amplitudes, but the linearised model acts as a linear filter that translates the white spectrum into a coloured noise spectrum of the resulting metabolite fluctuations (see Section 4.5).

Thermodynamics affects both the generation and transmission of noise. When modelling chemical noise by the chemical Langevin equation, each reaction rate $v^{*}$ contains an additive noise term with spectral density $S\left(\omega \right)∼Dg\left(\sqrt{|v_{l}^{*}|}\right)$. However, this formula does not apply to the net rate $v_{l}$, but to the one-way rates $v_{+}$ and $v_{-}$ (see Appendix A). In reactions near chemical equilibrium with a given net flux, these rates become large and contribute strongly to chemical noise. In contrast, strongly driven reactions produce less chemical noise and can serve as rectifiers for noise propagation (resembling a diode). The latter effect is captured by $R_{p}^{c}\left(\omega \right)$, which depends on the elasticities and therefore on thermodynamic forces. Hence, to model metabolic fluctuations, we just need to know the reaction elasticities, which can be obtained by STM. Figure 7 shows correlated metabolite fluctuations in our *E. coli* example model (see Figure 3). In the simulations, fast fluctuations are strongly damped far from their source reactions, while slow fluctuations can propagate further, causing network-wide flux and metabolite variations.

The effects of other fluctuating parameters (including enzyme level and external metabolite concentrations) can be computed similarly: their spectral density matrices S(ω) have to be known or guessed. Given the strong damping of chemical noise, the slow fluctuations caused by enzymes and the environment will dominate the observed variability in fluxes and metabolite levels. Pink noise, with fluctuations mostly at lower frequencies, may give rise to state fluctuations that look quasi-static. Finally, static covariance matrices can also be used to describe subjective uncertainty (for example, in models with known metabolic states but unknown kinetics). Likewise, temporal fluctuations may represent uncertainties or, as it were, information: for example, if we model a cell in an unknown fluctuating environment and if we interpret metabolic fluctuations as signals, the information they carry about the environment can be quantified by Shannon information and can be computed from correlations and autocorrelations of these variables.

### 2.5. Network Structure and Thermodynamic Forces Shape Metabolic Dynamics

The structure, dynamics, regulation, and function of metabolic systems are closely related. If proteins cooperate (e.g., if they belong to the same protein complex or pathway), this leads to synergisms and may be reflected in epistasis (i.e., synergisms of gene deletions towards a fitness-relevant variable), phylogenetic profiles [65], co-expression (or a temporal order of activation along the pathway [66]), and shared regulation mechanisms (e.g., between enzymes encoded in an operon). Similarly, network structure is also reflected in metabolite fluctuations and their correlations [67]. Of course, all these functional patterns portray network structure in indirect ways: while network edges are sparse, connecting adjacent metabolites and reactions, the resulting dynamic effects link elements across the entire network (but reflecting, for example, proximity within pathways). As an example of global effects, we may consider the propagation of metabolic fluctuations: as we saw above, chemical noise originates in each single reaction and propagates in the network. On their way through the network, high-frequency components are strongly damped (so they almost vanish far from their source reaction), while low-frequency components travel through the entire network, leading to slow or quasi-static variations of the flux distribution. To understand this in detail, we need to relate (local) network structure to (network-wide) metabolic dynamics. Once we understand causal dynamic effects (e.g., enzyme levels affecting metabolite concentrations and fluxes), we may invert this relationship and ask what enzyme profiles are best suited to achieve a desired metabolic behaviour [21,24,51,59], and what regulation mechanisms can realise these profiles.

How are network structure, dynamics, and function related precisely? To see this, we note that enzyme control and synergies, metabolite fluctuations, and possibly optimal differential expression patterns [24] reflect response coefficients, and that these coefficients depend on local network structure and local fluxes, concentrations, and reaction elasticities. Metabolic perturbations (of an enzyme, a metabolite concentration, or a reaction rate) have immediate local effects, which entail network-wide long-term effects. The immediate local dynamics is determined by the kinetics of single reactions at given metabolite levels, while the network-wide dynamics emerges from them as metabolite levels change dynamically after a perturbation. MCT describes these two stages, respectively, by elasticities and by control or response coefficients. Local network structure and kinetics are encoded in the stoichiometric matrix **N** and elasticity matrix **E**_c_, which is shaped itself by thermodynamics and enzyme saturation as considered in STM. The matrix product **N**
**E**_c_ yields the Jacobian matrix, which is local and sparse. The inverse of this Jacobian matrix leads to the non-sparse control and response matrices that describe network-wide changes of steady states. Another link between (local) network structure and (network-wide) metabolic behaviour can be made by the summation and connectivity theorems of MCT [18,20]: the control coefficients along a stationary flux distribution (which depend on network structure, but not on kinetics) must have a fixed sum (summation theorem), while the control coefficients around a metabolite are constrained by local elasticities (connectivity theorem). Hence, the control coefficients are constrained by ths network structure, but their precise values also depend on the elasticities—which can be explored by STM.

As we already saw, thermodynamic forces can have various effects on dynamics. Within a reaction, large forces imply a large energy dissipation per flux, and the relatively small one-way fluxes (given by $v_{+}\approx v,v_{-}\approx 0$) cause only little chemical noise. Due to the backward flux, the enzyme works efficiently and the enzyme demand per net flux is low. At the same time, the net rate is insensitive to the product concentration, so the product elasticity is low. Moreover, if a thermodynamic force is large, flux reversals would require large concentration changes, which makes such reversals physiologically difficult or impossible. As we saw before, due to their influence on elasticities, the thermodynamic forces have an impact on flux control in the entire network. Finally, thermodynamic forces, saturation values, and fluxes define typical relaxation times for metabolites. Local metabolite perturbations $\delta c_{i}$ tend to be dampened, as described by the linearised dynamics $\delta \dot{c_{i}}=\sum_{l}n_{il}E_{li}\delta c_{i}$. If the initial state is stable, the dynamics will decrease the initial perturbation, and $c_{i}$ will return to its reference value [68]. The relaxation time $\tau_{i}=1/\left(\sum_{l}n_{il}E_{li}\right)$ can be estimated from thermodynamic forces and enzyme saturation values.

In a linear pathway, these effects are easy to see: a strongly driven reaction has a low product elasticity and is not affected by downstream processes: it has full flux control and leaves no flux control to the downstream enzymes. Therefore, a change in downstream enzyme levels has no effect on the flux: it just changes the metabolite levels. Dynamically, the forward-driven reaction would act as a rectifier for metabolic fluctuations, just like a diode in electrical circuits: fluctuations can pass only in one direction. In reactions with finite thermodynamic forces, similar tendencies can be expected (see Appendix A). For example, due to their large flux control, such reactions are likely targets for regulation in linear pathways [47].

How can all this be shown using models? While fluxes in linear pathways can be described by a closed formula, no such formulae exist for fluxes in larger networks. Even if similar behaviour can be expected in larger networks, probably it cannot be proven analytically. However, STM provides the tools to study all this in detail.

## 3. Discussion

Let me summarise some main advantages of STM. First, it clarifies the links between metabolic concentrations and fluxes, enzyme parameters, and enzyme efficiencies and shows how these factors can be varied by cells. We can understand the interplay of enzyme concentrations, metabolite concentrations, and fluxes, simulate periodic random perturbations and chemical noise [26], and predict enzyme adaptation to changing supply and demand or to enzyme-inhibiting drugs [24]. Second, STM provides a link between flux analysis and kinetic models, showing how a metabolic state—for instance, fluxes and forces predicted by thermodynamic Flux Balance Analysis—determines the metabolic dynamics around it. Third, STM makes MCA applicable when model kinetics are missing and allows us to translate metabolic networks into kinetic models when data are scarce. This can also be useful to explore possibilities or variances in cell populations.

STM lends itself to ensemble modelling, model construction, and optimisation. In global variability analysis, model variables are sampled from broad distributions reflecting plausible parameter ranges [69,70,71]. For model construction, available kinetic, thermodynamic, or metabolic data [72,73] can be inserted into a model or be used to define ranges or distributions for sampling. While automatically constructed models are not fully reliable, they can be improved by drawing on existing, hand-curated models. STM can be used to convert given models with known kinetic parameters into standardised models with generic reversible rate laws. Conversely, after a model has been constructed by STM, some of the rate laws may be replaced by detailed laws obtained from enzyme assays [74]. In this case, the equilibrium constants of these rate laws should be imposed during STM. Finally, to provide existing pathway models with realistic dynamic boundaries, we may couple them to linearised and reduced models of the surrounding network [27], again constructed by STM.

To use STM for optimisation, we can replace our sampling steps by optimisation loops. In the dependence schema, the enzyme levels at predefined fluxes are convex functions of logarithmic metabolite concentrations and kinetic constants [75]. Convex optimality problems resembling Enzyme Cost Minimisation [47] can be defined and be solved, for example, with genetic algorithms. However, this should not be seen as a simulation of evolution because “mutations” and “selection” in these simulations do not act on enzyme levels as in reality, but directly on metabolite concentrations and fluxes. Alternatively, STM can also be a framework for parameter estimation. In a Bayesian setting resembling Model Balancing [75], finding the posterior mode would be an optimality problem with a “fitness function” encoding the log posterior. Posterior sampling could be implemented by applying Monte Carlo Markov Chain methods to the same problem. Whether we optimise a biological fitness or a posterior probability, and whether we use deterministic optimisation or probabilistic sampling methods, the dependence schemas of STM always provide a good parameterisation of our metabolic models.

In metabolic engineering, STM can be used in different ways. First, models may be built from available data by inserting known values or sample model variables around them. Model predictions may be used to choose promising pathway designs when data are scarce. Second, STM may reveal the effects and affordances of pathway features, e.g., the effects of structural variants, enzyme kinetics, or regulation. Third, we may use STM for optimisation with several objectives. For example, production yield (which only depends on the flux profile) and dynamic stability (which also depends on kinetics) can be subsequently optimised. In a cell, these objectives cannot be controlled separately: external perturbations will change fluxes and stability at the same time. However, STM calculation can tell us about possible trade-offs and about probabilities of finding favourable solutions, and can guide our search for suitable growth media or genetic modifications.

How can we predict the effects of enzyme perturbations such as single-enzyme inhibitions or differential expression of many enzymes? Metabolic responses may be quasi-static or dynamic and concern metabolite concentrations and fluxes. Constraint-based models ignore kinetics and replace them by heuristic rules: FBA predicts the most favourable flux changes, while MoMA predicts minimal changes. In both cases, enzyme repression can be modelled by restricting a reaction to smaller fluxes (knock-down) or zero fluxes (knock-out) and computing the perturbed flux distribution by maximising a metabolic objective (in FBA) or minimising the necessary changes in flux (in MoMA). If multiple enzymes are repressed, the flux in a linear pathway depends on the lowest flux bound while the other repressions have no effect. Therefore, in linear pathways, both methods predict buffering interactions. MCT, in contrast, predicts metabolic responses by kinetic models and control coefficients which reflect thermodynamics, enzyme saturation, and regulation and can be determined by STM. Reactions are not simply “reversible” or “irreversible”, like in FBA, but vary gradually between near-equilibrium and strongly forward driven reactions depending on thermodynamic forces. The quantitative effects of these forces, e.g., on the transmission of fluctuations, can be studied systematically. Moreover, an enzyme inhibition may change the control coefficients themselves, which leads to synergisms. MCT captures these synergisms by synergy coefficients.

Of course, MCT has its limitations. First, it holds only for small perturbations: with large perturbations, predicted concentrations may become negative. In steady-state MCT, this can be avoided by modelling log-concentrations. In dynamic simulations, we need to consider absolute concentrations, and to avoid negative concentrations heuristic correction functions can be applied. Second, STM assumes that enzyme levels are known and, by default, does not predict enzyme adaptations. In fact, the extra effects of such adaptations are second-order effects and can be neglected if perturbations are small (except for some systems at bifurcation points), or if we focus on dynamics much faster than enzyme changes. However, there are ways to consider enzyme adaptation in our predictions: if they are known from experiments, they can be inserted as enzyme perturbations; otherwise, optimal adaptation may be predicted based on a principle using control coefficients from STM [24].

STM does not only provide a framework for model construction, but it also clarifies physical and statistical dependencies between cell variables. A dependence schema, as in Figure 2, summarises constraints and shows how variables can be sampled, fitted, optimised, or chosen manually. To construct a dependence schema, we choose a set of physically independent “basic” model variables and treat all other variables as dependent. Along with strict physical dependencies, statistical dependencies can be defined by setting distributions for the basic variables, which lead to a joint distribution of all model variables. While a schema, endowed with a prior, determines all statistical dependencies between variables, the same dependencies may be expressed by different schemas, some of which allow us to use simple priors. Dependence schemas can be used for model construction, fitting, optimisation, and statistical analysis, and to learn how model variables determine metabolic behaviour. Knowing the model constraints allow us to discard infeasible models (e.g., describing a perpetuum mobile), and to restrict the model results, for example, the possible ranges of variables. By combining a dependence schema with probability distributions, plausible or measured data values can be inserted into the model and the remaining uncertainties can be assessed.

The dependencies and covariances of variables do not only matter in models, but also for cells in reality: kinetic constants or state variables may co-vary between the states of a cell, between cells in a population, during evolution, or between species. Some of this covariation is caused by physical constraints. For example, Haldane relationships between $k^{\mathrm{cat}}$ and $K_{M}$ limit the ways in which enzyme mutations can change these parameters [76]. In general, we do not know how enzyme mutations will change these parameters. Can we see some of them as independent and the others as dependent on them? For example, can we assume that Michaelis constants and forward $k^{\mathrm{cat}}$ values vary independently, and see backward $k^{\mathrm{cat}}$ as derived variables? Similarly, should we assume that enzyme concentrations correlate positively with catalytic constants (because both quantities are under a selection pressure for high enzyme capacity) or negatively (because if an enzyme is highly efficient, less of this enzyme is needed)? In fact, there is no “true” set of basic parameters in reality, only pragmatic choices in models: if variables have low statistical dependencies and large effects in the model, it is convenient to treat them as independent. Moreover, dependencies are always conditional: two variables may be dependent, but independent given another variable. In summary, dependencies between cell variables are important, and dependence schema are a good tool to describe them.

STM and SKM implement a retromodelling approach: we first choose a metabolic reference state and then the elasticities, rate laws, and control properties in this state. Retromodelling allows us to define state variables and kinetic constants separately and to study the effects on dynamics. For instance, by varying the thermodynamic forces we may turn reactions into rectifiers for metabolic fluctuations and adjust the stability and control of metabolic states. In unbranched pathways—where pathway fluxes can be computed analytically [77]—these effects are easy to understand. In larger networks, this is impossible, but STM allows us to study such effects both by sampling and by formulae that clarify the steps from thermodynamics to elasticities and from elasticities to control coefficients.

Retromodelling can be helpful for optimisation with several objectives, e.g., to achieve not only desired fluxes, but also desired control properties such as homeostasis, robustness, and adaptability. We begin with a desired flux distribution and construct, step by step, other state variables and kinetic constants that support this flux distribution. In each step, we can account for different constraints and objectives, e.g., optimal production rates when choosing the flux distribution. This “reverse” approach may help us think differently about optimisation in evolution or biotechnology, how metabolic states can be tuned, and with what protein costs. For example, we may imagine that cells need to optimise, first and foremost, their fluxes and concentrations under standard conditions, and secondly stabilise this state or make it homeostatic or better controllable by tuning the elasticities. To achieve this, evolution may vary thermodynamic and kinetic properties of reactions, tweak saturation values or add regulations, leading to different control patterns. For example, inhibiting an enzyme allosterically may stabilize the metabolic state, but will make the enzyme less efficient. To keep the fluxes unchanged, the lower enzyme efficiency must be compensated by higher enzyme levels. During model construction by STM, such targeted changes can easily be applied. This shows again that STM is not just a tool for model construction, but a framework for thinking about variability and optimality in cell populations, in evolution, or in engineered cells.

## 4. Materials and Methods

### 4.1. Constructing Kinetic Metabolic Models

Model building by STM relies on kinetic and thermodynamic laws and on Metabolic Control Theory. An overview of concepts and formulae is given in Appendix A. To translate metabolic networks into kinetic models, we need to integrate data about kinetics, thermodynamics, and metabolic states. In general, model construction poses a number of challenges: (i) finding realistic rate laws and kinetic constants; (ii) ensuring consistent equilibrium states of the model, with metabolite concentrations leading to vanishing fluxes; (iii) choosing a reference state with realistic fluxes and metabolite concentrations; and (iv) assessing variability and uncertainties in model parameters and metabolic states. A metabolic model can be parameterised automatically with the help of generic rate laws, including mass-action, power-law [78], linlog [79], thermodynamic-kinetic [80] or modular rate laws [46,70]. The use of reversible rate laws with consistent kinetic constants (i.e., constants satisfying Wegscheider conditions and Haldane relationships [81]) can guarantee consistent equilibrium states.

In kinetic models, enzyme levels and external metabolite concentrations are usually treated as parameters that determine the steady state. However, fitting such models to fluxes is difficult. Retromodelling reverts this procedure: we start from a flux distribution and construct metabolite concentrations and rate laws around it. We can do this as follows. After choosing the fluxes, the MDF method [47] can be used to identify reactions with poor thermodynamic forces and to choose metabolite concentrations that avoid such thermodynamic bottlenecks. Then, there are different ways to proceed. On the one hand, one may define reversible rate laws, choose plausible metabolite concentrations, and compute the enzyme demands [37,82,83], or optimise metabolite and enzyme concentrations simultaneously for a minimal enzyme cost [84]. On the other hand, one may first construct a metabolic state (including fluxes and metabolic concentrations) and then find kinetic constants that realise this state.

### 4.2. Elasticities and Their Dependence on Thermodynamic Forces

Enzyme kinetics and metabolic control are closely related to thermodynamic forces (Figure 1). In this section, I summarise some important formulae. The metabolic dynamics close to a reference state depend on (unscaled) reaction elasticities, defined as derivatives $E_{c_{i}}^{v_{l}}=\partial v_{l}/\partial c_{i}$ of kinetic rate laws $v_{l}\left(c\right)$ with respect to metabolite concentrations $c_{i}$. Elasticities describe the immediate, local effects of metabolite perturbations on reaction rates: to define them, we formally assume that metabolite concentrations are not dynamic variables, but parameters to be tuned from the outside. At high substrate concentrations, an enzyme becomes saturated and an additional substrate has little effect, so the elasticity is low. Elasticities between rates and other variables, e.g., enzyme levels, are defined accordingly. Importantly, the unscaled elasticities $E_{c_{i}}^{v_{l}}$ can be rewritten as $E_{c_{i}}^{v_{l}}=v_{l}{\hat{E}}_{c_{i}}^{v_{l}}\frac{1}{c_{i}}$ with unitless scaled elasticities defined as $E_{c_{i}}^{v_{l}}$ ${\hat{E}}_{c_{i}}^{v_{l}}=\frac{c_{i}}{v_{l}}\frac{\partial_{l}}{\partial c_{i}}$ (or ${\hat{E}}_{c_{i}}^{v_{l}}=\partial \ln\left|v_{l}\right|/\partial \ln c_{i}$). Scaled elasticities describe effective reaction orders. For irreversible rate laws, we obtain ${\hat{E}}_{c_{i}}^{v_{l}}\approx 1$ in the enzyme’s linear range and ${\hat{E}}_{c_{i}}^{v_{l}}\approx 0$ near saturation. If a reaction has two substrate molecules of the same chemical type, we obtain an elasticity ${\hat{E}}_{c_{i}}^{v_{l}}>1$. Formulae for second-order elasticities defined, via second derivatives are given in Appendix A. To linearise a metabolic model around a reference state, we only need the stoichiometric matrix and the unscaled elasticity matrix $E_{c}$. Both matrices can easily be obtained if the reference concentrations, fluxes, and rate laws are known. A perturbation (e.g., shifting the initial concentrations by a vector Δc(t)) will lead to a linearised dynamics; by solving the differential equations $\frac{d}{dt}\Delta c\left(t\right)=A\Delta c$ with the Jacobian matrix $A=NE_{c}$, we can simulate the propagation of dynamic perturbations across the network and determine their long-term effects [27]. In models with conserved moieties, a complication arises: the internal stoichiometric matrix has non-full row rank, and we need to use independent metabolite concentrations as the free variables. Their concentrations follow a dynamic equation $\frac{d}{dt}\Delta c^{\mathrm{ind}}\left(t\right)=A\Delta c^{\mathrm{ind}}$ with Jacobian matrix $A=N^{\mathrm{ind}}E_{c}L$, and $c(t)=L(c^{\mathrm{ind}}\left(t\right)-c^{\mathrm{ind}}\left(0\right))+c\left(0\right)$, where $N=LN^{\mathrm{ind}}$ and **N** has full row rank.

The choice of enzyme elasticities reflects some functional trade-offs. At low enzyme saturation, elasticities are high, enzyme efficiency is low, and the enzyme demand per flux (i.e., the inverse enzyme efficiency) is high. At high saturation, conversely, the reaction rate hardly changes with the substrate level, so the reaction is “stiff”: fluctuations in inflowing substrate cannot be buffered and lead to large fluctuations in substrate levels. Hence, the optimal choice of saturation values (and thus of elasticities) will reflect trade-offs between enzyme demand and favourable control properties.

If we know the fluxes and concentrations in a metabolic state, how can we find the elasticities? The elasticities depend on the rate laws with many unknown parameters. Modular rate laws (see Appendix A) are generic reversible rate laws based on simple enzyme mechanisms, with random-order binding. The formulae contain terms of the form $\beta_{li}^{X}=c_{i}/\left(c_{i}+k_{li}^{X}\right)$, called saturation values. A saturation value describes the saturation of an enzyme with a reactant or small-molecule regulator, where $c_{i}$ is a metabolite concentration, and $k_{li}^{X}$ is the dissociation constant between enzyme and metabolite (where X stands for M (reactants), A (activators), or I (inhibitors)). In writing *k*^M^ instead of $K_{M}$, the superscripts are used for convenience to leave space for lower reaction and metabolite indices, for example $k_{li}^{M}$. In modular rate laws, which rely on a quasi-equilibrium approximation [46], the $k^{X}$ are dissociation constants and half-saturation concentrations at the same time. Saturation values range between 0 and 1, and from saturation values and metabolite concentrations, we can reconstruct the dissociation constants $k_{li}^{X}$.

The scaled elasticities of modular rate laws consist of two terms [46],
(1)$$\begin{array}{ccc} {\hat{E}}_{c_{i}}^{v_{l}} & = & {\hat{E}}_{li}^{\mathrm{rev}}+{\hat{E}}_{li}^{\mathrm{sat}}. \end{array}$$

The thermodynamic “reversibility” term ${\hat{E}}_{li}^{\mathrm{rev}}$ depends directly on the thermodynamic force $\theta_{l}$, while the kinetic term ${\hat{E}}_{li}^{\mathrm{sat}}$ arises from kinetics and depends on the saturation values. Since the forces are coupled through Wegscheider conditions, all elasticities can be interdependent. However, given the forces and metabolite concentrations, the saturation values can be freely varied without violating any physical laws (for proof, see Appendix A). Details about modular rate laws and their first- and second-order elasticities can be found in [46].

All thermodynamically consistent reversible rate laws share the same numerator, which has the shape of mass-action rate law [46]. The numerator determines the thermodynamic elasticity term ${\hat{E}}_{li}^{\mathrm{rev}}$ [46]
(2)$$\begin{array}{c} {\hat{E}}_{li}^{\mathrm{rev}}=\frac{v_{+l}m_{li}^{S}-v_{-l}m_{li}^{P}}{v_{l}}=\frac{\zeta_{l}m_{li}^{S}-m_{li}^{P}}{\zeta_{l}-1} \end{array}$$
with substrate and product molecularities $m_{li}^{S}$ and $m_{li}^{P}$ and a flux ratio $\zeta_{l}=\frac{v_{+l}}{v_{-l}}=e^{\theta_{l}}$. The flux ratio, and therefore the elasticity term ${\hat{E}}_{li}^{\mathrm{rev}}$, depends on the thermodynamic force $\theta_{l}=-\Delta_{r}G_{l}/RT$, with Boltzmann’s gas constant *R* and absolute temperature *T*. For forward reactions near equilibrium (i.e., small positive thermodynamic forces $\theta_{l}\approx 0$), the substrate terms (where $m_{li}^{S}>0,m_{li}^{P}=0$) are close to infinity and the product terms (where $m_{li}^{S}=0,m_{li}^{P}>0$) are close to −∞. For completely forward-driven reactions (with thermodynamic force $\theta_{l}\to \infty$), we obtain elasticities $m_{li}^{S}$ for substrates and 0 for products. Between these extremes, the elasticities vary with the thermodynamic force. With a small force of 1 kJ/mol ≈ 0.4 RT and with molecularities and stoichometric coefficients equal to 1, Equation (Equation 2) yields scaled substrate elasticities of ≈3 and product elasticities of ≈−2. In contrast, for a force of 10 kJ/mol, we obtain values of 1.02 (substrate) and −0.02 (product).

The form of the kinetic elasticity term ${\hat{E}}_{li}^{\mathrm{sat}}$ depends on the type of modular rate law used. With mass-action or power-law rate laws without regulation, the kinetic term ${\hat{E}}_{li}^{\mathrm{sat}}$ vanishes and the elasticities follow directly from network structure and driving forces: (3)$$\begin{array}{c} \hat{E}={\hat{E}}^{\mathrm{rev}}=Dg{\left(v\right)}^{-1}\left[Dg\left(v_{+}\right)M^{S}-Dg\left(v_{-}\right)M^{P}\right]=Dg{\left(\zeta -\hat{1}\right)}^{-1}\left[Dg\left(\zeta \right)M^{S}-M^{P}\right]. \end{array}$$

With other rate laws, a kinetic elasticity term ${\hat{E}}_{c_{i}}^{v_{l}}$ needs to be added. For example, the simultaneous-binding modular (SM) rate law with non-competitive activation and inhibition [46] yields a kinetic term consisting of terms related to substrates S, products P, activators A, and inhibitors I: (4)$$\begin{array}{ccc} {\hat{E}}_{ci}^{vl} & = & {\hat{E}}_{li}^{\mathrm{rev}}-\underset{{\hat{E}}_{li}^{\mathrm{den}}}{\underbrace{\left[m_{li}^{S}\beta_{li}^{M}+m_{li}^{P}\beta_{li}^{M}\right]}}+\underset{{\hat{E}}_{li}^{\mathrm{reg}}}{\underbrace{\left[m_{li}^{A}\left(1-\beta_{li}^{A}\right)-m_{li}^{I}\beta_{li}^{I}\right]}}. \end{array}$$

The terms ${\hat{E}}_{li}^{\mathrm{reg}}$ and $-{\hat{E}}_{li}^{\mathrm{den}}$ reflect two parts of the rate law: a prefactor for regulation and the rate law denominator. Equation (Equation 4) shows that STM generalises Structural Kinetic Modelling: the elasticity formula in SKM,
(5)$$\begin{array}{ccc} {\hat{E}}_{c_{i}}^{v_{l}} & = & m_{li}^{S}\left(1-\beta_{li}^{M}\right)+m_{li}^{A}\left(1-\beta_{li}^{A}\right)-m_{li}^{I}\beta_{li}^{I}, \end{array}$$
is a limiting case of Equation (Equation 4) for completely forward-driven reactions, where $\beta_{li}^{M}=0$ and ${\hat{E}}_{li}^{\mathrm{rev}}=m_{li}^{S}$. We now see the effect of reversible reactions: by using Equation (Equation 4) (STM) instead of Equation (Equation 5), the elasticities become thermodynamically consistent and interdependent. With the common modular rate law [46] (or “convenience kinetics” [70]), the formula for scaled elasticities is more complicated: (6)$$\begin{array}{c} {\hat{E}}_{c_{j}}^{v_{l}}=\beta_{lj}\frac{\zeta_{l}m_{lj}^{S}-m_{lj}^{P}}{\zeta_{l}-1}-\beta_{lj}\frac{m_{lj}^{S}\psi_{l}^{+}+m_{lj}^{P}\psi_{l}^{-}}{\psi_{l}^{+}+\psi_{l}^{-}-1}+m_{li}^{A}\alpha_{li}^{A}-m_{li}^{I}\beta_{li}^{I}, \end{array}$$
where $\psi_{l}^{\pm }=\prod_{l}{\left(1+c_{i}/k_{li}^{M}\right)}^{m_{li}^{\pm }}$. Formulae for other rate laws and second-order elasticities can be found in [46] and Appendix A.

There is another useful elasticiy formula. In reactions with a positive flux, all reversible rate laws can be written in the factorised form [85]
(7)$$\begin{array}{c} v=ek^{\mathrm{cat}}\eta^{\mathrm{rev}}\left(\theta \right)\eta^{\mathrm{sat}}\left(c\right) \end{array}$$
where the efficiency terms are unitless numbers between 0 and 1 and the thermodynamic efficiency is given by $\eta^{\mathrm{rev}}\left(\theta \right)=1-\exp\left(-\theta \right)$. Here, forward $k^{\text{cat}}$ values and equilibrium constants appear as basic parameters, where $k^{\text{cat}}$, $k^{\text{eq}}$, and θ are defined with respect to the flux direction. Using the factorization (Equation 7) and assuming constant efficiency terms $\eta^{\mathrm{rev}}$ and $\eta^{\mathrm{sat}}$, we obtain a linear enzyme–flux relationship: the metabolite concentrations do not play a role and all elasticities vanish. A bit more realistically, we can compute $\eta^{\mathrm{rev}}$, in each state, from the known metabolite concentrations and values, thus considering the effect of driving forces (while still setting $\eta^{\mathrm{sat}}=$ const.), thus ignoring saturation or regulation effects. In the general case with variable efficiency terms, the formulae for scaled elasticities ${\hat{E}}_{c_{j}}^{v_{l}}=\frac{c_{i}}{v_{l}}\frac{\partial v_{l}}{\partial c_{i}}$ read (see Appendix A)
(8)$$\begin{array}{ccc} {\hat{E}}_{c_{j}}^{v_{l}} & = & \frac{-1}{e^{-\theta_{l}}-1}n_{il}+\frac{\partial \ln\eta_{l}^{\mathrm{sat}}}{\partial \ln c_{i}}, \end{array}$$
where the first term arises from $\eta^{\mathrm{rev}}\left(\theta \left(c\right)\right)$ and the second term depends on the choice of rate law. Without the second term (e.g., assuming full saturation), we obtain an elasticity formula that depends only on thermodynamic forces. Below we are interested in the interplay between thermodynamics and saturation, so this formula will not be considered any further.

### 4.3. Model Construction by STM

In STM, the mathematical structure of a model is described by a dependence schema (Figure 2a) that lists all independent “basic” variables and the remaining “derived” variables. For brevity, I will use the term “variables” not only for fluxes, metabolite concentrations and enzyme concentrations, but also for kinetic and thermodynamic constants. Following the schema, we can build models systematically step by step (Figure 2b): we start from a network (reaction stoichiometries and regulation arrows), determine the state variables (concentrations, fluxes, equilibrium constants), choose the saturation values, and compute consistent (first- and second-order) elasticities by using Equation (Equation 1). The kinetic constants can be reconstructed easily. From the second-order elasticities, we obtain second-order control and response coefficients, describing enzyme synergies for static [23] or periodic perturbations [26]. Any feasible model with the type of rate laws considered can be obtained in this way (for details, see Appendix A).

During model construction, model variables can be chosen in various ways: they can be replaced by data values, sampled, fitted, or optimised. Below, we assume that fluxes are fitted to data or determined by FBA, metabolite concentrations are chosen within feasible ranges and in line with the flux directions, and saturation values are sampled. With this set-up, the hardest step is the choice of fluxes, which need to be thermo-physiologically feasible; that is, thermodynamically realisable with metabolite concentrations in given physiological ranges, which also implies that the flux distribution must be loopless [86]. Such fluxes can be obtained by EBA or thermodynamic FBA (based on mixed-integer linear problems) or by starting from given fluxes and removing infeasible loops. Notably, STM can be applied to non-steady reference states, for example to flux data that were mapped to a pathway model without all necessary in- and outfluxes. However, in this case, the next step—Metabolic Control Theory—will require a steady reference state. To construct such a state with measured fluxes, models may be augmented with extra incoming and outgoing reactions. Given the fluxes, feasible metabolite concentrations can be determined, e.g., by the Max-Min Driving Force (MDF) method [47]. If a flux distribution is thermodynamically infeasible, some thermodynamic forces will go against the flux direction. To apply STM in this case, we may change the forces by adding extra terms attributed to hypothetical extra metabolites (see Appendix A). To guarantee stable states in our model ensemble, we may proceed like in SKM: for each model instance, we check that the model has a stable reference state (i.e., that the eigenvalues of the Jacobian matrix have negative real parts), and discard all models with unstable states. In the resulting ensemble, the model variables may show different distributions and correlations: e.g., saturation values may not follow the original distributions and become correlated. It is not generally known how flux patterns, metabolite concentrations, thermodynamic forces, and enzyme saturation contribute to stability and how the chances of finding a stable state depend on network size. This would be an interesting question to be studied by STM.

As noted before, in model construction by STM the basic model variables can be sampled, freely chosen, fitted to data, or optimised for a fitness objective. We can use all this to incorporate extra knowledge or data. A consistent set of kinetic constants, as input data for STM, can be obtained by parameter balancing. Parameter balancing is a method for translating incomplete, inconsistent, and uncertain data into complete, consistent sets of model variables [52,54]. Given such input data, known *k*^M^ values can be converted into saturation values. These values can then be inserted directly, or saturation values can be sampled around them (uniformly or following a beta distribution) to account for uncertainties or missing information [87]. Known *k*^cat^ values can be used similarly: in the dependence schema, the *k*^V^ values, together with equilibrium constants and fluxes, determine the turnover rates $k_{+}^{\mathrm{cat}}$ and $k_{-}^{\mathrm{cat}}$ and the enzyme concentrations *e* for the same reaction. To match them to data, we can first choose *k*^V^ values at random and then modify them to obtain a good fit of forward *k*^cat^ value, enzyme concentration, and flux; or we use a different schema with forward *k*^cat^ values instead of *k*^V^ as basic variables.

When applying STM, I noticed that reference metabolite concentrations and fluxes—and not just the saturation values—are really important for model dynamics because of how they shape the unscaled elasticities. For example, the known metabolite concentrations make the reconstructed threonine pathway model (in Appendix A) work rather well. Hence, if metabolite concentrations and fluxes are unknown, one should always sample them broadly together with the saturation values. I also observed that, for obtaining reasonable model dynamics, small driving forces should be avoided (as assumed in the MDF method): ideally, in the reference states, driving forces should be restricted to values θ>1.

Finally, STM may also be used for Bayesian model fitting and to predict biologically optimal states. In these cases, model variables are not sampled but treated as choice variables to be fitted or optimised. Unfortunately, the resulting optimality problems are typically non-convex and hard to solve. Model balancing, which defines simplified convex optimality problems, may be used instead [75].

By repeatedly sampling model parameters, we obtain an ensemble of models that all share the same structure but show different parameter values [22,28,33]. Model ensembles can impose a network structure and some other choices made by the modeller (e.g., given fluxes and metabolite concentrations), but allow for variability in all other variables. Different model assumptions or model variants will lead to different model ensembles. To study how choices of model structure, metabolic state, or kinetics determine metabolic dynamics, we generate model variants, translate them into model ensembles, and search for significant differences in their behaviour. Significance tests are described in Appendix A.

The STM algorithm can be modified in various ways.

Cell growth. To model metabolism in growing cells, the formulae must be adapted. For balanced growth at a cell growth rate λ, all cellular compounds must be reproduced continuously, so the fluxes follow a mass balance condition Nv−λc=0 with an extra dilution term. Metabolite concentrations and fluxes are tightly coupled by this equation, and in model construction they must be chosen together. In the kinetic model, a dilution term $-\lambda c_{i}$ must be added to the ODE of each metabolite, and the Jacobian matrix (which also appears in formulae for the control matrices) contains an extra term −λI. Moreover, conserved moieties in such models must always vanish because otherwise they would be diluted, thus preventing a steady state.Kinetic constants as basic variables. Instead of choosing saturation values and concentrations and then computing the constants $k_{li}^{X}$, we may treat the constants $k_{li}^{X}$ as basic variables and compute the saturation values from them. The distributions for ratios $c/k^{M}$ and for saturation values $\beta =\frac{c/k^{M}}{1+c/k^{M}}=\frac{c}{k^{M}+c}$ are obviously related. If β is uniformly distributed in ]0,1[, the ratio $c/k^{M}$ shows a probability density function $\mathrm{prob}\left(c/k^{M}\right)=\frac{1}{{\left(1+c/k^{M}\right)}^{2}}$, i.e., $\ln(c/k^{M})$ follows a logistic distribution with location parameter 0 and scale parameter 1 (see Appendix A). In practice, we may conveniently sample saturation values from uniform or beta distributions, while kinetic constants may be sampled from sufficiently narrow gamma distributions or from similar log-normal distributions (for the choice of probability distributions, see Appendix A). By sampling saturation values not within ]0,1[ but in a smaller range, one may avoid full saturation, and one may sample $k^{M}$, $k^{A}$, and $k^{I}$ values around known experimental values.Multiple steady states. To build a model with multiple steady states, in the metabolic state phase we choose one set of equilibrium constants, but several sets of concentrations and fluxes for the different steady states; in the kinetics phase the constants $k_{li}^{M},k_{li}^{A}$, and $k_{li}^{I}$ and velocity constants $k_{l}^{V}$ (geometric means of forward and backward catalytic constants) are chosen or sampled, for example, by parameter balancing [37,52,54]. Finally, enzyme concentrations for each state are computed by matching reaction rates from the rate laws to predefined fluxes.

Eventually, for more realistic models, some of the generic rate laws may be replaced by more detailed rate laws obtained from enzyme assays [74]. For other extensions and modifications of STM, see Appendix A.

### 4.4. Metabolic Control and Synergy Effects

If enzymes are perturbed by inhibition or transcriptionally, how will this change the metabolic concentrations and fluxes? Inhibiting an enzyme makes substrates accumulate and products deplete. These changes have further effects, which counteract the original effect and eventually lead to steady-state changes in the entire network. To understand such indirect effects, let us think of a simple example: a reaction perturbed by a single parameter (e.g., an enzyme level or an external substrate concentration). In MCT, the direct effects on reaction rates are described by parameter elasticities and the further effects (on steady-state metabolite concentrations and fluxes) are described by control coefficients. By multiplying parameter elasticity and control coefficient, we obtain the response coefficient (the sensitivity between parameter perturbation and metabolite concentrations or fluxes). Small perturbations can be treated as additive: the effects of simultaneous perturbations can simply be summed over. A variation δe of enzyme activities, at fixed external metabolite concentrations ($\delta c^{\mathrm{ext}}=0$), will lead to a flux variation $\delta v=C^{V}E_{c}\delta e$, with the unscaled control coefficient matrix $C^{V}=I-E_{c}L{\left(N_{R}E_{c}L\right)}^{-1}N^{\mathrm{ind}}$. Increasing an enzyme activity can increase or decrease fluxes across the network. Each matrix column describes the (positive or negative) effects of an enzyme on all the fluxes. The control coefficients are usually unknown, but STM allows us to guess them based on information about network structure, reference fluxes, metabolite concentrations, thermodynamic forces, and enzyme saturation.

For small enzyme perturbations, the effects of enzyme perturbations on fluxes and other target variables can be described by a linear approximation: a variation $\delta e_{j}$ of the enzyme level leads to flux changes $\delta v_{l}=R_{e_{j}}^{v_{l}}\delta e_{j}$. For larger perturbations, however, the results may be different, and we describe this as synergy effects: an enzyme perturbation changes not only the target variable, but also the very effects of enzymes on this variable. The extra effect is called synergism, or “antagonism” when it is negative. Antagonisms arise, for example, if two enzymes share the same substrate: as one enzyme concentration goes down, the substrate concentration increases and the flux catalysed by the other enzyme goes up, increasing its own flux control. Synergisms do not only exist between enzymes, but between any variables (or discrete network features) that affect the reaction rates. Besides enzyme concentrations, this may include enzyme inhibition, knock-outs, differential expression, or perturbations of external metabolite concentrations. Generally, to quantify synergisms, we consider two perturbations (e.g., of enzyme activities) and a target variable (e.g., the biomass production rate). If single perturbations a and b change the target value by factors $w_{a}$ or $w_{b}$ (typically smaller than 1, if the target variable is a maximisation objective), we may expect, as a guess, that a double perturbation will lead to a change $w_{a}\cdot w_{b}$. We now compare this guess to the actual change $w_{ab}$. If the two values differ, we describe this by a synergy effect $\eta_{\mathrm{ab}}^{z}=\ln\frac{w_{\mathrm{ab}}}{w_{a}w_{b}}$. Otherwise, if we expect additive instead of multiplicative changes, the difference $\eta_{\mathrm{ab}}^{z}=w_{\mathrm{ab}}-\left(w_{a}+w_{b}\right)$ can be used to define a synergy effect.

How can synergisms be predicted from models? We need a model that predicts changes in a metabolic target variable (e.g., a flux or metabolite concentration) after single or double perturbations of enzyme concentrations (in kinetic models) or of fluxes (in flux analysis). Synergies in kinetic models can be computed numerically: we vary two enzyme levels and simulate the effect on the steady-state fluxes. A first inhibition will change the flux control coefficients of all enzymes, and therefore the effect of a second inhibition. For small perturbations, the synergy effect can be approximated to second order. While the usual first-order response coefficients capture linear effects of single enzymes, the second-order response coefficients (or “synergy coefficients”) capture synergisms of enzyme pairs and second-order effects of single enzymes. Concentrations can be described on log scale to guarantee that concentrations stay positive. Logarithms applied to fluxes would make it impossible for predicted fluxes to change their direction. With logarithmic enzyme changes $\Delta \ln e_{a}$ and $\Delta \ln e_{b}$ (where $\Delta \ln e_{a}\approx \frac{\Delta e_{a}}{e_{a}}$), the synergism is given by ${\hat{R}}_{\mathrm{ab}}^{z}\approx {\hat{E}}_{e_{a}e_{b}}^{z}\Delta \ln e_{a}\Delta \ln e_{b}$, where ${\hat{R}}_{e_{a}e_{b}}^{z}$ is the scaled synergy coefficient. If two enzymes are inhibited, negative synergisms (${\hat{R}}_{e_{a}e_{b}}^{z}<0$) are called aggravating while positive synergisms (${\hat{R}}_{e_{a}e_{b}}^{z}>0$) are called buffering. The second-order effects of single-enzyme perturbations are described by self-synergisms: they are usually aggravating because inhibition tends to increase an enzyme’s own control, which increases the effect of the inhibition on the target variable.

Different modelling frameworks make different assumptions. FBA tries to maximise the metabolic objective even after a perturbation; MoMA assumes that fluxes show minimal changes despite perturbations, thus behaving homeostatically; MCT instead predicts flux responses from metabolic dynamics, assuming that enzyme concentrations remain constant (as considered below) or are optimally adapted (see [24]). How are the synergisms computed in practice? In FBA or kinetic model simulations, synergy effects are simulated one by one for each enzyme pair. To model flux perturbations by FBA, we first solve an FBA problem with a given objective function (e.g., biomass production flux) to obtain a flux profile and a resulting target variable (typically, the objective itself). To mimic an enzyme inhibition, we limit the catalysed flux, constraining it to a certain percentage of the original flux (e.g., 90% for a small perturbation, 50% for a large perturbation or knock-down, 0% for complete inhibition or knock-out). By solving the FBA problem again, we obtain the perturbed target value, and by repeating this procedure for single and double perturbations, we can compute all synergy effects. Synergy effects in MoMA are computed similarly: again, the inhibited reactions are constrained, but the new fluxes are supposed to resemble as much as possible the unperturbed flux. With synergy effects computed like this, a double inhibition in a linear pathway has a simple effect: the new pathway flux is the minimum of the two inhibited fluxes. Thus, after a first inhibition, the second inhibition has either its full effect or no effect at all. In both cases, we obtain buffering synergisms. In MCT, synergisms are obtained from second-order elasticities, which can be sampled by STM (see Appendix A). Epistasis—synergy effects of gene deletions on cell fitness—can be predicted similarly, considering gene knock-outs as large perturbations and cell growth as the target variable. To obtain an epistasis measure suited for FBA, with comparable ranges for positive and negative epistasis, Segrè et al. [88] introduced a correction for buffering interactions (see Appendix A). With MCT, this is not necessary because positive and negative synergisms are already in similar ranges.

### 4.5. Variability and Fluctuations in Cells

How are the variability, correlations, and temporal fluctuations of cell variables shaped by network structure, fluxes, and thermodynamic forces? For simplicity, let us first consider static variability between cells and then dynamic fluctuations within a cell—keeping in mind that static variability is just a a limiting case of dynamic fluctuations at frequency zero. To describe a cell population, we may use a single model for all cells, but with differences in the input variables that determine the steady states (e.g., randomly distributed enzyme concentrations $e_{l}$ with a covariance matrix $\Sigma_{e}$). In a first-order approximation, the covariance matrix of metabolite concentrations is given by $\sum_{c}=R_{e}^{c}\sum_{e}R_{e}^{c^{⊤}}$, with the response coefficient matrix $R_{e}^{c}$ [22]. A more precise second-order approximation requires synergy coefficients. Similar formulae hold for all perturbation parameters (e.g., external metabolite concentrations, kinetic constants), and for variations in fluxes. The same covariance formula can also be used to describe subjective uncertainty, e.g., about predicted steady states, based on uncertain model parameters [22].

Fluctuations in time are described similarly (see [26] and Appendix A). Tracing stochastic fluctuations is difficult, but statistical properties such as amplitudes, time correlations, and correlations between fluctuating variables are easy to obtain in linearised models. We just need to consider their Fourier transforms, that is, the noise spectrum in frequency space. Fluctuating parameters and variables are characterised by their spectral power density matrices, which resemble the static covariance matrices **Σ**, but are complex-valued (Hermitian instead of symmetric) and frequency-dependent. The spectral power density S(ω) describes the “weight” of at frequency ω in the stochastic process. Correlated fluctuating variables are described by a matrix with diagonal elements describing the spectral power densities and off-diagonal matrix elements describing correlations. For uncorrelated white noise, this matrix is an identity matrix S(ω)=I. If the spectral densities of the original parameter fluctuations are known, the resulting spectral densities of state variables can be computed in a linear approximation (see Appendix A). In linear metabolic models with a stable reference state, noise parameters $p_{j}$ with spectral density matrix $S_{p}$ will lead to fluctuating metabolite concentrations with a spectral density matrix $S_{c}\left(\omega \right){=}_{}^{}\left(\omega \right)R_{p}^{c}\left(\omega \right)S_{p}(\omega )R_{p}^{c}{\left(\omega \right)}^{⊤}$. Again, diagonal elements are real-valued and contain the spectral density of a concentration $c_{i}$, while correlations are described by complex-valued off-diagonal elements. Random distributions of steady-state fluxes are described similarly by a matrix $S_{v}\left(\omega \right){=}_{}^{}\left(\omega \right)R_{p}^{v}\left(\omega \right)S_{p}(\omega )R_{p}^{v}{\left(\omega \right)}^{⊤}$. The effects of parameter fluctuations depend on the frequency: slow fluctuations (that is, $S_{}\left(\omega \right)\approx 0$ except for ω≈0) have quasi-static effects, creating permanent differences between cells.

What can we learn from the spectral densities about measured variability in cells? In measurement data, noise appears as a variability on a particular time scale, the time resolved in our measurement. If concentration measurements take a second, faster fluctuations will be averaged over and our data can reveal variations only on slower time scales. Hence, what interests us is the variability in a variable, averaged over a time window Δt (or after applying to our stochastic process a smoothing kernel of width Δt). Such results will be shown in Figure 7 in Results, centre bottom (for calculation details, see Appendix A).

## Figures and Tables

**Figure 1 metabolites-12-00434-f001:**
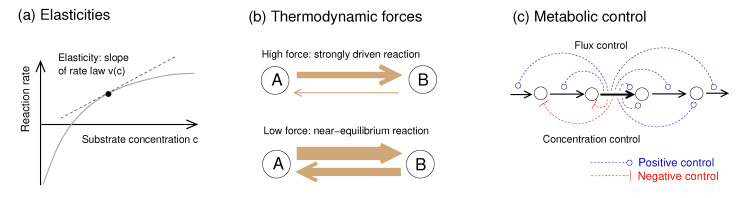
Metabolic rate laws and dynamics: elasticities, thermodynamic forces, and metabolic control. (**a**) A reaction rate depends on metabolite concentrations as described by a rate law v(e,c). The slope ∂v/∂c is called reaction elasticity. (**b**) In strongly driven reactions, with a driving force $\theta =\ln\frac{v_{+}}{v_{-}}\gg 1$, the net rate is dominated by the forward rate and the product elasticity is almost zero. In contrast, close to chemical equilibrium (with a driving force θ≈0 and large forward and backward rates, but a small net rate), the scaled elasticities ${\hat{E}}_{c_{i}}^{v_{l}}=\frac{c_{i}}{v_{l}}\frac{\partial v_{l}}{\partial c_{i}}$ are large. (**c**) Metabolic control coefficients describe how perturbations in single reactions shape steady-state fluxes and metabolite concentrations across the network. If an enzyme is inhibited or repressed, upstream metabolites accumulate and downstream metabolites deplete. The details of this response depend on network structure, flux distribution, and reaction elasticities. Thermodynamic forces shape metabolic control via the elasticities: a strongly driven reaction, with its low product elasticity, is insensitive to downstream processes and deprives all downstream enzymes of their flux control.

**Figure 2 metabolites-12-00434-f002:**
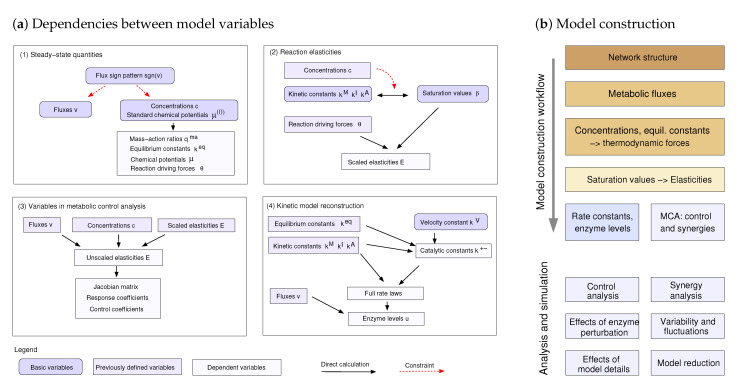
Dependence schema and systematic model construction. (**a**) Dependencies between model variables (kinetic constants and state variables) in kinetic metabolic models. A dependence schema describes physical or logical dependencies between variables and can serve as a blueprint for model construction. For simplicity, the schema is shown in four parts, corresponding to four steps of model construction. (1) Metabolic state phase. Fluxes and thermodynamic forces must have the same signs. Predefined flux directions define sign constraints on fluxes and thermodynamic forces, and fluxes and chemical potentials can be sampled under these constraints. (2) Kinetics phase. Saturation values $\beta_{li}^{M}$, $\beta_{li}^{A}$, and $\beta_{li}^{I}$ can be chosen independently between 0 and 1. Together with metabolite concentrations and thermodynamic forces, they determine kinetic constants ($k_{li}^{M}$, $k_{li}^{A}$, and $k_{li}^{I}$) and reaction elasticities. The elasticities further determine control properties (3) as well as kinetic constants and enzyme concentrations (4), allowing us to reconstruct the entire kinetic model. In the graphics, some variables stem from previous steps (types of variables marked by colours). (**b**) Model construction around a metabolic reference state. Based on a dependence schema, basic model variables can be freely chosen or sampled, while derived variables are computed from them.

**Figure 3 metabolites-12-00434-f003:**
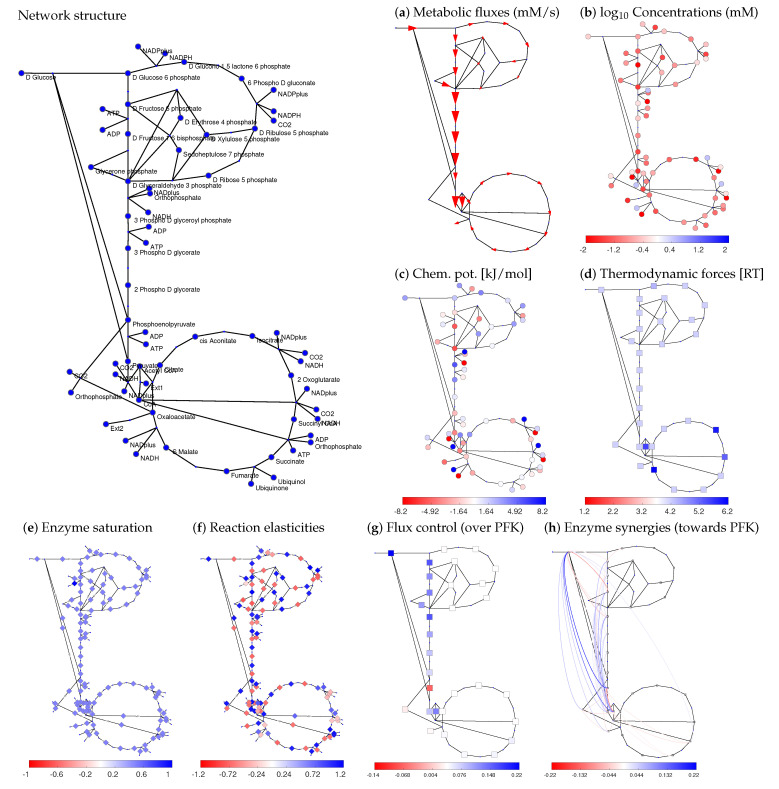
Systematic model construction by STM. Model of central carbon metabolism in *Escherichia coli* (for model details, see Data Availability). Network structure and flux data (from cells respiring on glucose) were taken from [51]. The panels show different types of variables obtained during model construction following the schema in Figure 2 (circles: metabolites; arrows and squares: reactions). By repeatedly sampling the basic variables and computing the others, a model ensemble can be constructed. (**a**) Thermodynamically feasible fluxes (red arrows) obtained by flux minimisation (data from [53]); (**b**) metabolite log-concentrations. Metabolite concentrations and thermodynamic forces were determined by thermodynamic balancing [54] of metabolite and reaction Gibbs free energy data. (**c**) chemical potentials; (**d**) thermodynamic forces in units of RT; (**e**) saturation values, set to standard values of $\frac{1}{2}$. Alternatively, the saturation values could be determined from data or be sampled at random between 0 and 1; (**f**) scaled reaction elasticities ${\hat{E}}_{c_{i}}^{v_{l}}$; (**g**) scaled control coefficients $\hat{(}{C)}_{v_{l}}^{r_{\mathrm{PFK}}}$ for the flux in upper glycolysis (phosphofructokinase reaction) as the target variable; (**h**) enzyme synergies (given by scaled second-order control coefficients) for the glycolytic flux. Positive values are shown in blue, negative values in red, zero values in white. For clarity, only enzyme synergies in the outer 5 percent quantiles are shown.

**Figure 4 metabolites-12-00434-f004:**
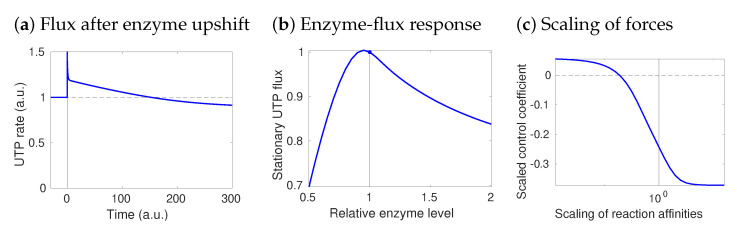
Paradoxical flux control: a model of UTP rephosphorylation in human hepatocytes. Fluxes in central metabolism were obtained in [53] by FBA with UTP production as the objective. Here, flux control coefficients were obtained by STM with standard elasticities, i.e., saturation values of 0.5. Paradoxically, the UTP-regenerating enzyme NDK has a negative control over its own flux. (**a**) Metabolic dynamics after an NDK upshift. A higher NDK activity first speeds up the reaction, but later the rate drops below its initial value. (**b**) Dose–response curve between NDK amount and steady-state UTP production. In the reference state (vertical line), the curve slope (response coefficient) is negative: an enzyme increase decreases the flux. (**c**) The response coefficient depends on thermodynamic forces. To see this, all thermodynamic forces were increased by a common factor (*x*-axis): with higher thermodynamic forces, the response (*y*-axis) becomes more negative.

**Figure 5 metabolites-12-00434-f005:**
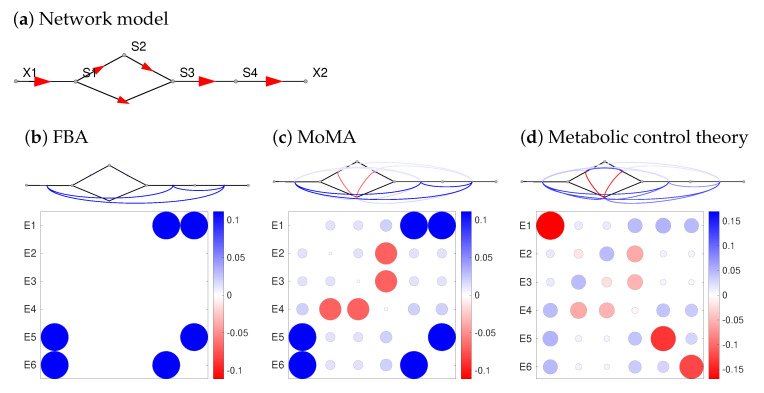
Enzyme synergy effects in a linear pathway with alternative routes. The (scaled) synergisms towards a fitness-relevant objective (e.g., biomass production) can tell us about epistasis. (**a**) network structure. A conversion from S${}_{1}$ to S${}_{3}$ can occur directly or via the intermediate S${}_{2}$. The following plots show synergy effects for double inhibitions predicted by FBA (**b**) or MoMA (**c**). To simulate enzyme inhibitions, a flux decrease to 90 percent of the original value was imposed by setting a bound at 0.9 times the unperturbed reaction flux. The double inhibition of an enzyme is simulated by applying the inhibition twice, i.e., leading to a relative flux decrease factor of 0.81. Synergy effects are shown by line colours (red: aggravating, blue: buffering) and as a matrix. Colour ranges span the observed synergy effects in each panel (red: negative; white: zero; blue: positive). Small values, below one percent of the maximal absolute value, are not shown. (**d**) Synergy effects computed by Metabolic Control Theory, assuming common modular rate laws and half-saturated enzymes. A model ensemble with saturation values sampled from uniform distributions yields almost the same results on average. Also a different rate law (the simultaneous binding modular rate law) yields very similar results (see Appendix A). Compare Figure 3 for synergy effects in *E. coli*.

**Figure 6 metabolites-12-00434-f006:**
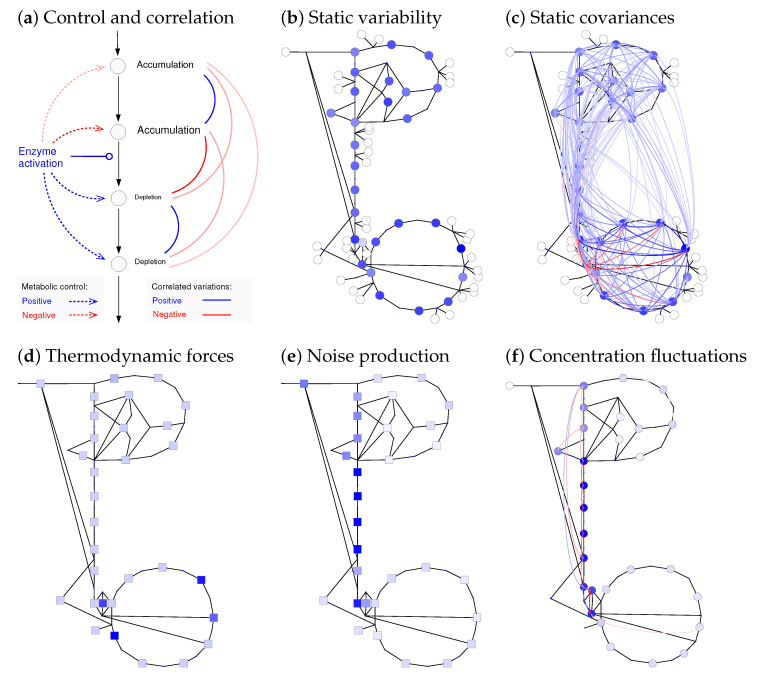
Static changes and dynamic fluctuations of metabolite concentrations. (**a**) Variability of metabolic states caused by perturbed reaction rates (schematic drawing). An enzyme activation (left) increases a single reaction rate (solid arrow), which changes the steady-state metabolite concentrations and fluxes (dotted arrows). Similarly, variability in enzyme activities (quasi-static random changes) causes slow correlated metabolite variability (curved lines on the right, blue: positive; red: negative); (**b**) variation of metabolite concentrations in *E. coli* central metabolism (Figure 3) caused by uncorrelated variable enzyme concentrations (geometric standard deviation of enzyme levels: 2). Standard deviations of log concentrations are shown by colours; (**c**) correlated metabolite variations (covariances of log concentrations) are shown as coloured lines (values below 10% of the maximal value were removed for clarity). Metabolite variances (in (**b**)) and covariances (in (**c**)) computed from first-order response coefficients. Local enzyme fluctuations lead to network-wide metabolite fluctuations: the frequency spectra are related by spectral response coefficients. (**d**) Thermodynamic forces (same data as in Figure 3d). Effects of chemical noise in a cell volume of 2 μm${}^{3}$ and with a glycolytic flux of 1 mM/min; (**e**) reactions close to equilibrium (small thermodynamic forces) produce strong chemical noise because of their large forward and backward fluxes. Spectral power density of the original noise; (**f**) resulting metabolic fluctuations: fast fluctuations at a frequency of 1 s. Formulae are given in Appendix A.

**Figure 7 metabolites-12-00434-f007:**
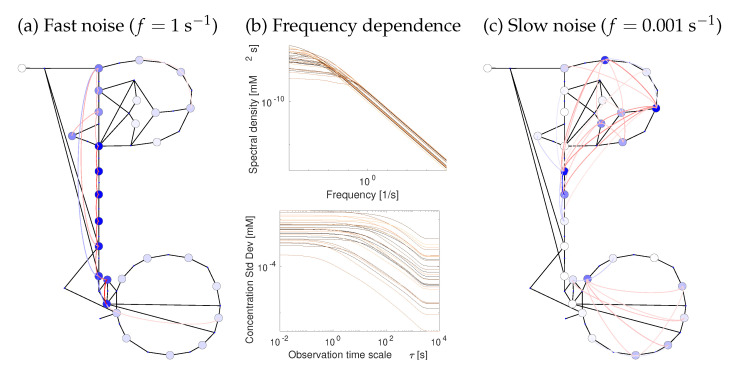
Metabolite fluctuations due to chemical noise. Chemical noise causes random fluctuations of metabolite concentrations and fluxes. In a chemical Langevin equation, chemical noise is modelled by adding white-noise terms to the reaction rates. Fast and slow fluctuations are damped differently in the network. (**a**) High-frequency metabolite fluctuations (spectral densities at oscillation freuency *f* = 1 s^−1^) exist only close to the noise source. Circle colors show the spectral densities at 1 s^−1^, line colors show covariances (blue: positive; red: negative). (**b**) Metabolite fluctuations decrease with the frequency (see Figure 6). (**Top**): spectral densities, each curve corresponds to the variance of one metabolite over a range of frequencies. (**Bottom**): each curve corresponds to the standard deviation of a metabolite concentration (square root of spectral density, shown on the *y*-axis), computed for different time resolutions of observation (*x*-axis). For details, see Appendix A. (**c**) Low-frequency metabolite fluctuations (oscillation period of 17 min) are correlated along the entire pathway. Results for flux fluctuations are shown in Appendix A. Smoothing at different time resolutions changes the variance of metabolite fluctuations. At low time resolutions, high-frequency fluctuations are filtered out (see Appendix A).

## Data Availability

Matlab code for kinetic models, elasticities, and STM, using the sparse tensor toolbox [91,92] is freely available at github.com/liebermeister/stm (accessed on 25 April 2022). The STM algorithm is described in Appendix A. All parameter settings can be found in the Matlab scripts. SBML and SBtab formats are used for models and data. The *E. coli* model, a modified version of the model from [51], is included and described.

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
