# Peer review of "Structural Thermokinetic Modelling"

_metabolites, 2022, doi:10.3390/metabo12050434_

Round 1

Reviewer 1 Report

Reviewer report: Structural thermokinetic modelling

Metabolic networks, even at the genome scale size, are nowadays frequently build based on the annotations available from the corresponding genome. Even if they have proven its usefulness in several areas and particularly in metabolic engineering, improving and expanding their range of applications requires to consider additional kinetic and thermodynamic constrains. Due to the size of these kind of models, the necessary information to build a complete deterministic kinetic model is usually lacking. In this paper the author proposes a new method, namely Structural Thermokinetic Modelling (STM), addressing most of the problems arising in previous approaches such as those of Structural Kinetic Modelling (SKM).    

After introducing the problems and limitations of previous approaches, the paper describes the method as an improvement of the SKM by including thermodynamics and therefore avoiding possible inconsistencies. A workflow for building realistic kinetic models is described, assuring to obtain consistent models either using generic reversible rate laws or realistic rate laws derived from enzyme assays and automatic parameterization. It starts by selecting basic or independent ‘variables’ and systematically advancing to a full system description using a retromodelling approach. The method is flexible enough so that basic variables can be fitted to experimental data as well as sampled or optimized from a fitness objective. Possible variations of the method are described.

In a second part of the paper, different applications are showcased using mainly data from an E.coli bibliographic example. Thus, the effect of changing enzyme concentrations, their epistatic or synergy effects, the effect of noise or perturbations or the combined influence of structure, dynamics and thermodynamics on function are explored with examples, often using the approach of building an ensemble of models and analyzing their averaged properties. Thus, providing thoughtful reflections on the integrated operation of metabolic networks.

More detailed explanations of the methods, results or relationship with other approaches such as metabolic control analysis, are described in the accompanying supplementary materials. In addition, the software to perform those calculations is freely available in Github and I successfully tested the operation of the demonstration example.

As extensively shown in the text and the accompanying materials the proposed approach can be used for model fitting, analysis or optimization using experimental data or building model ensembles.

Thus, it is the opinion of this reviewer that it represents a significant advance from previous approaches and deserves publication. Only minor corrections in the text seem to be necessary as described below with a proposed correction:

Main text:

Line 128: ‘rate is does’-> rate does

Line 137: ‘’can  by independently’-> ‘can be independently’

Line 290: ‘In the case’ -> ‘In this case’

Line 457, 460a and 460b ‘epistatis’ -> epistasis’

Line 480: ‘an probability’ -> ‘a probability’

Line 505: ‘In model’-> ‘in the model’

Line 527: ‘figure ??’ a figure number is missing

Figure 3: ‘Grey arrows’  -> ‘red arrows’

Line 552: ‘SI section ??’ -> section number is missing

Line 595: : ‘SI section ??’ -> section number is missing

Figure 4: ‘Flux control coefficients (blue: positive; pink: negative)’ -> this does not appear in this figure

Figure 6 (a): if enzyme inhibition increases accumulation, the metabolic control should be positive. Is it not? Color of the arrows indicate the contrary. Please review.

Line 750: ‘separataly’ -> ‘separately’.

Line 767: ‘On the contrary’ -> ‘In addition’ or ‘analogously’

 Line 781: ‘circuite’ -> ‘circuit’

Line 837: ‘adaptationsin’ -> ‘adaptations in’

Line 855: ‘determines’ -> ‘determine’

Line 877: ‘a third variables’ -> ‘a third variable’

In Supplementary Information:

Line 204: ‘saxme’ -> ‘same’.

Line 309: ‘teh’ ->’the’

Line 462: ‘und’ -> ’and’

Line 466: ‘STM to be modified’ -> does not match very well please review for clarity

Line 511: ‘the distribution our’ -> ‘the distribution of our’

Line 539, 542: ‘epistatis’ -> epistasis’

Line 542: ‘In th eopposite’-> ‘In the opposite’

Line 698: ‘brackets’ -> ‘parenthesis’

Reviewer 2 Report

Well written, technically and structurally.  

The manuscript by Liebermerster proposed structural thermokinetic modelling to answer some of the critical question, for instance, translating metabolic networks into dynamic models. This work is quite important since thermodynamics can be used to obtain more precise predictions of flux control, enzyme synergies, correlated flux and metabolite variations.

The manuscript addresses important limitations of current flux model. The author proposed STM, explained very well of model construction, STM algorithm and flux results. This work can help us understand the interplay of enzyme concentrations, metabolite concentrations, and fluxes. This manuscript is of great deal of interest for the readers of Metabolism and metabolic flux analysis.

Reviewer 3 Report

This manuscript presents STM, an extension of SKM that considers thermodynamic constraints in the random sampling of kinetic parameters to generate model ensembles.

The manuscript represents extensive theoretical work that is very useful to better (more accurately or more consistently with physical/chemical constraints) model biochemical systems where available data is not sufficient for dynamical model parameterization (which is most often the case).    

The manuscript is written as a tutorial to STM implementation and examples and demonstrations of its usefulness.

I consider that the manuscript deserves to be published after minor revisions to correct typos.

Eventually, I would suggest to add a rational for using beta distributions, or log-normal for the sampling of parameter values.

Several typos that I detected are: (there are eventually more, please revise carefully)

line 192: Eˆvlci ≈ 0 in the linear range and Eˆvlci ≈ 1 near saturation : should it be opposite?

line 505: strange sentence, please revise

line 520: superscripts have been used previously, so the presented explanation should be moved to the first time they are used.

line 527, 553, 595: ??

line 750: separataly
